# Frailty as a Predictor of Outcomes in Subarachnoid Hemorrhage: A Systematic Review and Meta-Analysis

**DOI:** 10.3390/brainsci13101498

**Published:** 2023-10-23

**Authors:** Michael Fortunato, Fangyi Lin, Anaz Uddin, Galadu Subah, Rohan Patel, Eric Feldstein, Aiden Lui, Jose Dominguez, Matthew Merckling, Patricia Xu, Matthew McIntyre, Chirag Gandhi, Fawaz Al-Mufti

**Affiliations:** 1Department of Neurology, Westchester Medical Center, New York Medical College, Valhalla, NY 10595, USA; michael.fortunato@wmchealth.org (M.F.); flin3@student.touro.edu (F.L.); auddin@student.nymc.edu (A.U.); rpatel38@student.nymc.edu (R.P.); klui@student.touro.edu (A.L.); pxu@student.nymc.edu (P.X.); chirag.gandhi@wmchealth.org (C.G.); 2Brain and Spine Institute, Westchester Medical Center, New York Medical College, Valhalla, NY 10595, USAeric.feldstein@wmchealth.org (E.F.); jose.dominguez@wmchealth.org (J.D.); 3Department of Neurological Surgery, Oregon Health & Science University, Portland, OR 97239, USA

**Keywords:** subarachnoid hemorrhage, frailty, outcomes

## Abstract

Frailty is an emerging concept in clinical practice used to predict outcomes and dictate treatment algorithms. Frail patients, especially older adults, are at higher risk for adverse outcomes. Aneurysmal subarachnoid hemorrhage (aSAH) is a neurosurgical emergency associated with high morbidity and mortality rates that have previously been shown to correlate with frailty. However, the relationship between treatment selection and post-treatment outcomes in frail aSAH patients is not established. We conducted a meta-analysis of the relevant literature in accordance with PRISMA guidelines. We searched PubMed, Embase, Web of Science, and Google Scholar using “Subarachnoid hemorrhage AND frailty” and “subarachnoid hemorrhage AND frail” as search terms. Data on cohort age, frailty measurements, clinical grading systems, and post-treatment outcomes were extracted. Of 74 studies identified, four studies were included, with a total of 64,668 patients. Percent frailty was 30.4% under a random-effects model in all aSAH patients (*p* < 0.001). Overall mortality rate of aSAH patients was 11.7% when using a random-effects model (*p* < 0.001). There was no significant difference in mortality rate between frail and non-frail aSAH patients, but this analysis only included two studies and should be interpreted cautiously. Age and clinical grading, rather than frailty, independently predicted outcomes and mortality in aSAH patients.

## 1. Introduction

Frailty is a complex condition characterized by decreased physiological reserve across multiple organ systems, rendering individuals more susceptible to both endogenous and exogenous stressors and increasing the risk of adverse health outcomes [1]. Although there is no single cause responsible for frailty, it is likely due to the interaction of accumulated age-related deficits and chronic disease states across the lifespan, producing a vicious cycle that disrupts the normal aging process [2,3]. Frailty is considered dynamic and potentially reversible, and individuals are characterized as existing on a continuum between fit and frail [4]. Key to the concept of frailty is that it is a distinct entity from either disability (i.e., physical limitations that impede performance of one or more activities of daily living (ADLs)) or multimorbidity (i.e., the coexistence of two or more unrelated chronic disease states); it rather implies a state of “pre-disability,” although frailty, disability, and multimorbidity can all co-exist within the same individual [5].

The natural course of frailty is markedly heterogeneous, although two large-scale cohort studies—the Longitudinal Aging Study Amsterdam (LASA) and the InCHIANTI study—have shown a common progression of physical manifestations of physical decline, beginning with exhaustion, then slowed gait, decreased physical activity, and weakness [6]. Additionally, evidence from previous studies has demonstrated fairly consistent trends in frailty, such as increasing prevalence of frailty with older age, a significantly higher prevalence of frailty among women compared to men, and higher prevalence among certain racial and ethnic groups, including African Americans and Hispanics [7,8,9,10]. A recent systematic review of 23 studies analyzing risk factors and protective factors for frailty found statistically significant associations between frailty and sociodemographic factors, psychological factors, biological factors, lifestyle factors, and physical factors [11].

In the last two decades, the clinical understanding of frailty has expanded greatly; however, there is still no consensus among researchers and providers on a singular definition of frailty or the optimal way of assessing it. As a result of these discrepancies, there exist many different methods of quantifying frailty, with hundreds of scales developed in the last two decades alone. Among them, three of the most widely used frailty scales are the physical frailty phenotype (PFP), the modified frailty index (mFI), and the Gill Frailty Measure [12,13,14]. Of these, the majority of previous studies have defined frailty using the clinical criteria laid out by Fried et al. (2001), also known as the Frailty Phenotype, including exhaustion, unintentional weight loss, decreased physical activity, slowed gait, and poor grip strength [13]. The mFI explores both cognitive ability and comorbidities, while the other indices focus on physiological strength [15]. Regardless of the measurement tool, research has shown that frailty is a strong predictor for perioperative mortality and post-operative complications across various surgical interventions [16,17,18].

Frailty is quickly becoming a global health concern, as the burden of a rapidly expanding elderly population poses a serious challenge to healthcare systems worldwide, given this population’s elevated incidence of falls, hospitalizations, iatrogenic conditions, and earlier mortality. A deeper understanding of frailty is also desperately needed in order to facilitate a means of determining the most high-quality, cost-effective care for the elderly population and alleviating increased strain on the healthcare system. Though many methods for the management of frailty are currently available, such as protein-calorie supplementation, increased physical activity, and reduced polypharmacy, the evidence for successful interventions thus far is lacking [19]. Frail individuals are also more likely to experience unmet healthcare needs, an issue that is exacerbated by this dearth of evidence-based interventions at both the individual and system-wide level for managing and combating this chronic condition. Furthermore, frailty can augment the classical risk factors associated with normal aging, thus contributing to worse health outcomes. For example, frailty has been found to influence the relationship between Alzheimer’s disease pathology and the clinical manifestations of Alzheimer’s disease, such that frail Alzheimer’s patients were found to have a lower disease burden on pathology compared to their non-frail counterparts [20]. Thus, consideration of the frailty status of patients undergoing surgical treatment, especially among the elderly, is paramount to improving outcomes within this population.

Aneurysmal subarachnoid hemorrhage (aSAH), which constitutes approximately 85% of nontraumatic or spontaneous aSAH cases, is a critical neurosurgical emergency linked to substantial morbidity and mortality. It most commonly results from the rupture of an intracranial aneurysm, leading to bleeding into the subarachnoid space. The classic and most common presentation of aneurysmal aSAH is the “thunderclap headache”, which individuals will often call the “worst headache of their life” and is sudden, excruciatingly painful, and at maximum intensity from onset [21]. Other notable symptoms of aSAH include loss of consciousness, vomiting, meningismus, seizures, focal cranial nerve deficits, and hemiparesis [22]. The severity of aSAH is assessed using either the Hunt–Hess scale (grades 1–5) or the World Federation of Neurological Surgeons scale (grades 1–5). As with frailty, there is a significantly greater preponderance of aSAH cases in females, with one recent study finding 13.1 cases of aSAH in females per 100,000 versus 9.6 cases per 100,000 in men [23].

Treatment for aSAH can be conservative or surgical. Conservative treatment involves stabilization and acute management of life-threatening symptoms, monitoring, lowering of elevated intracranial pressure via head-of-bed elevation or medically, using agents such as mannitol, and prevention of vasospasm. Surgical treatment for aSAH includes endovascular interventions, such as surgical clipping of endovascular coiling, and other methods of reducing increased intracranial pressure, like hemicraniectomy. Surgical intervention is the only definitive treatment for aneurysm repair. However, surgery of any kind for aSAH entails inherent risks; hence, more conservative management is generally preferred, especially in patients with worse severity grading on admission. This has previously been attributed to medical nihilism on the part of providers, although recent advances in neurocritical care have called into question this paradigm [24]. The concept of frailty has not been fully explored within the aSAH cohort, and current research is mixed as to whether frailty is an effective predictor of outcomes [25,26,27]. It is thus unknown whether frail patients who experience aSAH would benefit more from surgical treatment, potentially leading to better outcomes than conservative management in select patients, or if the frailty label precludes further surgical consideration. An understanding of frailty within the aSAH population could potentially lead to a new avenue for stratifying preoperative risk for aneurysm repair. As such, we sought to conduct a systematic review and meta-analysis to examine the current literature regarding frailty within the aSAH population and evaluate the effect of frailty on mortality and outcomes in aSAH.

## 2. Materials and Methods

### 2.1. Search Strategy and Study Eligibility

A literature search was performed using PubMed, Embase, Google Scholar, and Web of Science in accordance with the Preferred Reporting Items for Systematic Review and Meta-analysis (PRISMA) guidelines using the search terms “subarachnoid hemorrhage AND frailty” and “subarachnoid hemorrhage AND frail”. Retrospective chart reviews, database reviews, and prospective studies that were written in English, peer-reviewed, and published from 1990 to 2022 were included. There were no restrictions on the length of the follow-up period. Studies were first screened through review of abstracts, then a full-article review was conducted. Studies lacking frailty measurements or relevance to the focus of this review were excluded (Figure 1). This study was not registered.

### 2.2. Data Extraction

Data on cohort age, frailty measurements, post-treatment outcomes, and clinical grading systems of aSAH were extracted. These systems included the Hunt–Hess scale (HH), modified Fisher scale (mFS), and World Federation of Neurological Surgeons (WFNS) scale. Other variables included country of origin, study type, and year of publication.

### 2.3. Meta-Analysis: Synthesis of Results

Relevant data from studies of interest were extracted, reorganized, combined, subcategorized, and otherwise cleaned as necessary prior to statistical analysis. In situations where data or variables needed for subcategorical analysis were not reported, studies were excluded. In calculating average frailty percentage across relevant studies, primary cohort and validation cohort data from one of the studies were combined [27]. In another study using temporal muscle thickness as a frailty measurement, frailty was defined as an Evans index (EI) ≥ 0.3 [29].

Meta-analysis was executed for average age, frailty percentage, and average modified Fisher scale (mFS) and mortality. Statistical analyses of these variables under a random-effect model were conducted using “Comprehensive Meta Analysis v. 3.3.070”. Due to bias and overestimation of heterogeneity by the point I^2^ estimate in small meta-analyses, confidence intervals were reported alongside the I^2^ statistic as suggested by von Hippel [30].

### 2.4. Quality Evaluation

We evaluated all included studies for their quality using the 2011 Oxford Centre for Evidence-based Medicine (OCEBM) levels of evidence [31]. In this framework, the studies with the lowest level of evidence (5) are based on expert opinion and mechanistic evidence, while the highest (1) are based on a systematic review of randomized controlled trials. 

## 3. Results

### 3.1. Search Results

The initial literature search using PubMed, Embase, Web of Science, and Google Scholar databases yielded a total of 74 results with keywords “subarachnoid hemorrhage AND frailty” or “subarachnoid hemorrhage AND frail”. After removing duplicate articles, 16 remained. After applying eligibility criteria and excluding studies not pertinent to our focus, we identified four studies that matched all of our inclusion criteria.

### 3.2. Demographics

Studies were conducted in three different countries: China, Japan, and the USA. One study included both a retrospective and a prospective cohort, while the others were all retrospective analyses [27]. All studies except one were conducted at a single center in the past 10 years [32]. For patient characteristics, only one study investigated patients ≥60 years old, while the others had no restrictions on age [27]. Using a random-effects model on the data from all four studies shows that the average age of the aSAH patients included was 57.8 years old (CI = 55.9–59.7, *p* < 0.001; Figure 2).

### 3.3. Frailty Measurements and Outcomes

Among the four studies included, three used the mFI as the frailty measurement, while one study used temporal muscle thickness as the frailty measurement [26,27,29,32] (Table 1). Percent frailty was found to be 30.4% (CI = 22.4–39.8%, *p* < 0.001) under a random-effects model in all aSAH patients (Figure 3).

Clinical assessments such as the mFS grade, HH score, and WFNS grade were used across the studies (Table 2). Based on our model, we can expect the average mFS score to be 3.3 (CI = 2.4–4.1, Figure 4). Unfortunately, statistical meta-analysis could not be performed for HH scores or WFNS grade due to differences in reporting style between studies.

All four studies reported mortality, but there were significant differences in how it was reported: one study reported mortality only after treatment, another study reported mortality both before and after, and the other two studies did not include such temporal information [26,27]. One study reported on mortality as part of poor functional outcomes using the modified Rankin scale (mRS), which could not be included in the meta-analysis since it was not an exclusive measurement of mortality [33]. Only one study reported in-hospital mortality, since the NIS database did not include follow-up information [32]. The overall mortality rate of patients with aSAH was found to be 11.7% when using a random-effect model (CI 7.4–18.1%, *p* < 0.001). There was no significant difference in mortality rate among frail versus. non-frail aSAH patients, but this analysis only included two studies and should be interpreted with caution (Figure 5a–c).

## 4. Discussion

In this meta-analysis of the effects of frailty on outcomes in aSAH patients, we identified 74 articles, four of which met our inclusion criteria, and determined that age and clinical status were better predictors of long-term outcomes than frailty in aSAH. The average age of these patients was 57.8 years old, which was similar to the mean age of aSAH patients in the general population [33,34,35]. Epidemiological studies of aSAH have shown the incidence of aSAH peaks at 50–60 years of age and then plateaus after 60 years old [36]. Incidence studies of frailty are exceedingly rare, with most studies only reporting prevalence within specific populations. Thus, from the literature, it has been shown that the prevalence of frailty increases with age; however, Collard et al. (2012) previously showed that this prevalence varies tremendously among community-dwelling older persons, between 4.0–59.1%, and was highest in individuals ≥ 85 years of age [7]. Of note, the prevalence of frailty appears to vary based on the manner of assessment; for example, when only physical frailty is assessed, the prevalence varies between 4.0 and 17.0%, but when the assessment includes physical, psychological, and social frailty, the prevalence varies between 4.2 and 59.1% [37]. Furthermore, there appears to be a geographic variation in the prevalence of frailty, with several studies identifying decreased prevalence in more affluent countries and increased prevalence in poorer countries [38]. Regardless, most cases of frailty begin by age 65 years, although it is important to note that not all individuals will develop frailty, and it is by no means an inevitability of advanced age [39]. Additionally, although possibly reversible, the likelihood of other disease states emerging also increases once frailty is established and allowed to continue without intervention, leading to potentially irreversible functional decline [40].

In aSAH patients, advanced age has previously been established as a major predictor for increased mortality and poor clinical outcomes [41,42]. Researchers have proposed that advanced age may denote a multifactorial process of impaired metabolism, slowed repair mechanisms, and loss of vascular autoregulation, thus rendering elderly patients more vulnerable to worse functional outcomes and increased mortality after the insult [41,42,43]. In fact, three studies in our analysis either revealed that frailty was not an independent predictor or highlighted age and clinical grades as more important factors [26,29,32]. Interestingly, in one study, researchers developed a prognostic model for predicting post-treatment outcomes in aSAH. They assigned a score of three for advanced age (≥80 years) compared to a score of one for frailty, illustrating that age is a more significant predictor of outcomes in aSAH prognosis than frailty [27]. However, given that increasing age is a prominent component of the frailty phenotype, it becomes difficult to disentangle the synchronous, likely synergistic, effects that age and frailty exert on the individual experiencing aSAH.

Multiple studies have also identified that scores on clinical grading scales, such as the WFNS, HH, or Glasgow Coma Scale (GCS), are predictive of outcomes in aSAH. Patients with a worse clinical grade on admission are associated with unfavorable functional outcomes at 3- and 12-month follow-up [44]. Other studies have also identified that early clinical grade changes or clinical grades after neurological resuscitation are more predictive of outcomes such as mortality [45,46]. Unfortunately, due to the heterogeneity in reporting styles among the papers included, we were unable to conduct a meta-analysis regarding clinical assessments.

In addition to age and clinical assessments, frailty is also a popular measurement in predicting outcomes in aSAH. Frailty is a relatively common phenotype in aSAH, as evidenced by our finding that 30.4% of aSAH patients were frail. In this meta-analysis, all included studies examined the correlation between frailty and short-term outcomes, including mortality, discharge location, functional outcomes, and complications. Specifically, two studies suggested frailty was an independent predictor of unfavorable outcomes [27,32]. Another recent study (not included in this meta-analysis) found that increasing frailty was linked to an increase in the incidence of complications, such as vasospasm, hydrocephalus, cerebral infarction, and acute respiratory failure, as well as increased length of stay and total cost, in patients undergoing endovascular repair of a ruptured intracranial aneurysm [47]. Researchers have proposed that since frail patients have decreased physiological reserve and suffer from more comorbidities, they may be less resistant to tissue damage following aSAH. Therefore, frailty may serve as a potential tool for treatment selection. In the cohort of patients with low-grade aSAH (HH score I-III), it has been reported that frail patients have a decreased chance of receiving interventions [26].

Currently, there are few studies focusing on using frailty as a clinical determinant in neurological conditions, and further studies are still warranted to investigate its utility. Outside of neurological fields, interventions to reverse frailty, especially physical frailty, include physical exercise, nutritional supplements, cognitive training, or a combination of these methods [48]. For example, in pulmonary transplant patients, most patients had either reversal or significantly improved frailty phenotype after transplantation and a 3-month pulmonary rehabilitation program. However, it is unclear whether such improvement was due to disease-specific intervention or a general improvement in physiological reserve [49]. The comprehensive geriatric assessment (CGA) has been proposed as a possible intervention to identify and prevent the development of frailty. One randomized, controlled trial utilizing a 6-month program incorporating the CGA found that this intervention reduced ADL disability by one-third and shortened the average length of stay in nursing homes by one week [50]. Further systematic reviews and meta-analyses of programs utilizing the CGA have demonstrated significant improvements in outcomes of frail older individuals across multiple settings [51,52,53,54].

This meta-analysis has several limitations. Only four studies were able to be included. Additionally, these studies represented only a limited number of countries (USA, Japan, and China), thus preventing wider generalizations about the frail populations of other nations from being drawn. Because of the limited number of studies included, publication bias may distort the mortality rate seen in aSAH patients. Through publication bias analysis, a slight asymmetry in the funnel plot is seen for standard error (SE) by Logit Mortality rate for frail and non-frail patients with aSAH (Appendix A). This could indicate a potential publication bias present in the studies included in this meta-analysis. In small meta-analyses (<7 studies), the point I^2^ statistic can overestimate heterogeneity. Although I^2^ was less than 50 (I^2^ = 46.9), results should still be interpreted cautiously and with potential heterogeneity in mind because Q > Q (df). Additionally, as suggested by the OCEBM criteria, all studies included are classified as having low evidence, since most cohorts were recruited retrospectively and a limited number of subjects were included. Furthermore, due to differences in study designs, we were unable to conduct a meta-analysis on some clinical grading scores and mortality. Hence, there is increasing need for more prospective studies to investigate the relationship between frailty and outcomes in aSAH.

The development of frailty is multifactorial in etiology and includes cognitive impairments, lack of regular exercise, chronic diseases, and poor nutrition. Increased emphasis should be placed on interventions that prevent the development of frailty by addressing these underlying causes. Walston et al. (2018) discussed four major types of interventions that have been implemented in order to improve the health of frail patients: exercise, nutrition modification, multicomponent interventions, and individual-specific geriatric care models [55]. The large number of recent systematic reviews and meta-analyses concerning frailty confirms the growing importance of this issue. These studies have determined key associations between frailty and a host of other detrimental chronic conditions, including hypertension, depression, heart failure, polypharmacy, and multimorbidity [56,57,58,59,60]. Furthermore, many of these studies highlight the dynamic, reciprocal interplay of these interactions, in which each condition is associated with an increased incidence of the other. However, as most of these studies are retrospective, future longitudinal, prospective studies are necessary, and well as studies determining whether a decrease in any of the associated conditions results in a consequent decrease in the incidence of frailty.

The study of frailty is hindered by inconsistencies and discrepancies at each point of measurement, complicating the endeavor tremendously. The current instruments used to assess frailty, though good for providing predictive value in recognizing which patients are at greatest risk for adverse outcomes, are not equipped to inform clinical practice guidelines or aid in the development of interventions to prevent or treat frailty. Moreover, the agreement between frailty instruments is highly variable, likely due to a fundamental disagreement on a singular definition of what frailty encompasses [61]. In addition to developing a standardized definition of frailty, future directions should focus on rectifying differences among the disparate frailty scales through the creation of one universal, standardized scale that takes into consideration all relevant concepts of frailty and provides measurable benefit in clinical practice. Finally, another recent focus in the study of frailty involves the use of biomarkers to more effectively diagnose frailty in older individuals. However, as with creating a single comprehensive frailty index, there is disagreement over selecting the appropriate frailty biomarkers. Those that have been proposed include measurements of musculoskeletal changes, stem cell changes, and hormonal changes, in addition to serum (hemoglobin, antioxidants), metabolic (hemoglobin A1c), and inflammatory markers (CRP, IL-6, TNF-α) [62]. What is agreed upon, however, is that no single biomarker by itself, including laboratory biomarkers, can encompass all the elements of frailty [63]. Future directions in the study of biomarkers in frailty will likely focus on multidimensional and multivariate analyses, which are better able to capture the complex interplay between different pathophysiological domains contained within this multifaceted etiology [64]. A recent meta-analysis of 80 studies involving 33,160 older individuals aged 60–88 years found shared biomarkers between frailty and sarcopenia, including albumin and hemoglobin, which could one day allow for monitoring of the frailty phenotype and aid in clinical decision-making [65].

While the specific type of treatment for each patient was not explicitly reported in our paper, it is indeed an important factor that can affect the interpretation of our findings. In our ongoing research, we plan to explore this aspect more comprehensively. Specifically, we intend to categorize the treatment modalities in a more granular manner, considering conservative, surgical, and endovascular approaches, and analyze their respective impacts on patients’ outcomes in the context of frailty. By addressing the specific treatment modalities and their interactions with frailty, we aim to provide a more nuanced and comprehensive understanding of how different treatments may affect aSAH patients with varying levels of frailty.

In this meta-analysis of four studies examining frailty measurements in aSAH patients, we show that frailty is a composite measurement that is interconnected with age and clinical manifestations. Age-related vascular changes can reduce the cerebral reserve, making older patients less resistant to neurological insults such as aSAH and more prone to worse neurological outcomes in the aftermath of a potentially devastating injury [43]. Since the current tools for measuring frailty are unable to capture cerebral reserve, age and clinical grading scales, rather than frailty, should be prioritized in the setting of aSAH in predicting patient outcomes.

## 5. Conclusions

Of the 74 studies pooled, 4 studies fit the criteria for inclusion in this meta-analysis. There are multiple frailty measurements used in assessing the aSAH population. Frailty is associated with unfavorable short-term outcomes but may not serve as an independent factor in prognosis and outcomes. Meanwhile, age and clinical gradings are important factors with regard to outcome and mortality. Further studies are needed to compare different frailty measurements in the aSAH cohort and investigate the influence of frailty in treatment selections.

## Figures and Tables

**Figure 1 brainsci-13-01498-f001:**
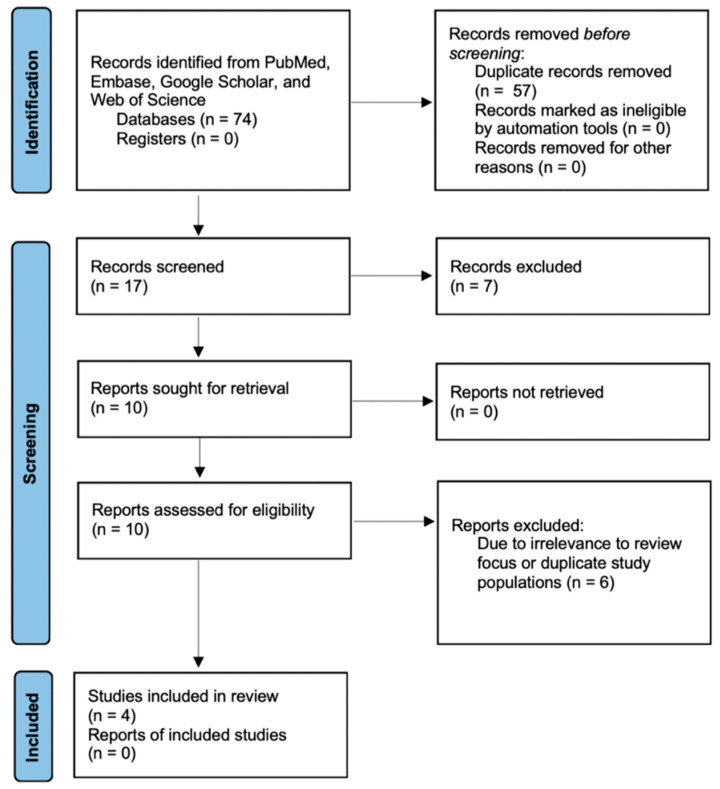
PRISMA study flow diagram demonstrating the number of studies included at each stage of data extraction. Adapted from the PRISMA 2020 flow diagram [28].

**Figure 2 brainsci-13-01498-f002:**
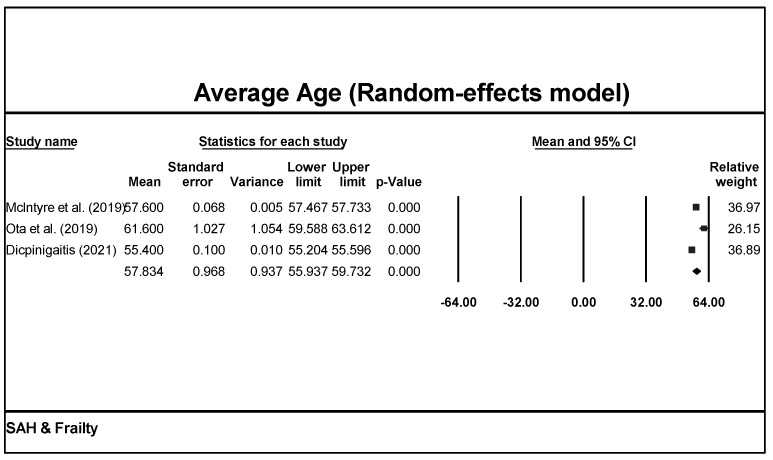
Meta-analysis of patient age of all studies revealed an average of 57.8 years under a random-effect model in aSAH patients [26,29,32].

**Figure 3 brainsci-13-01498-f003:**
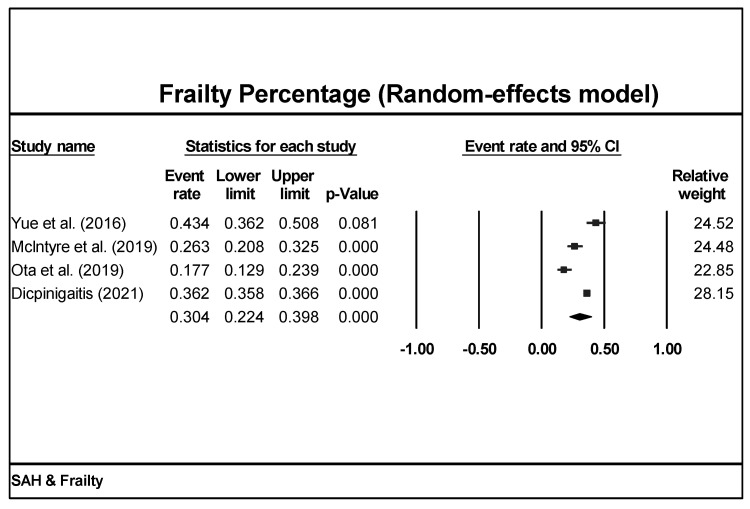
Meta-analysis of frailty percentage of all studies revealed that 30.4% of aSAH populations are frail [26,27,29,32].

**Figure 4 brainsci-13-01498-f004:**
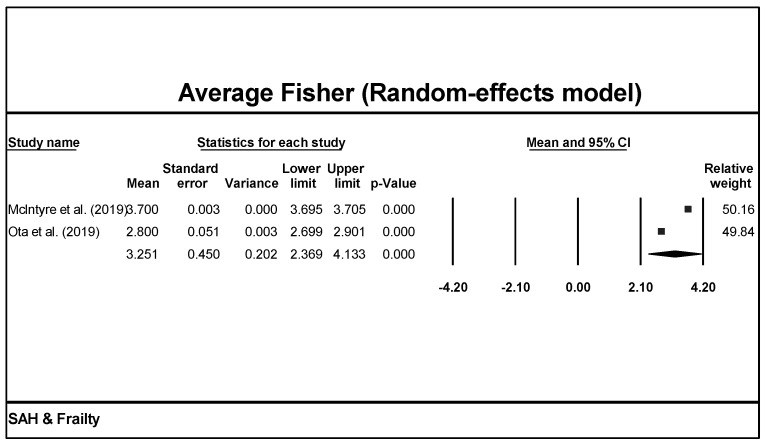
The meta-analysis of Fisher scores from the included studies indicated that among the aSAH populations where Fisher scores were reported, the average Fisher score was 3.3 [26,29].

**Figure 5 brainsci-13-01498-f005:**
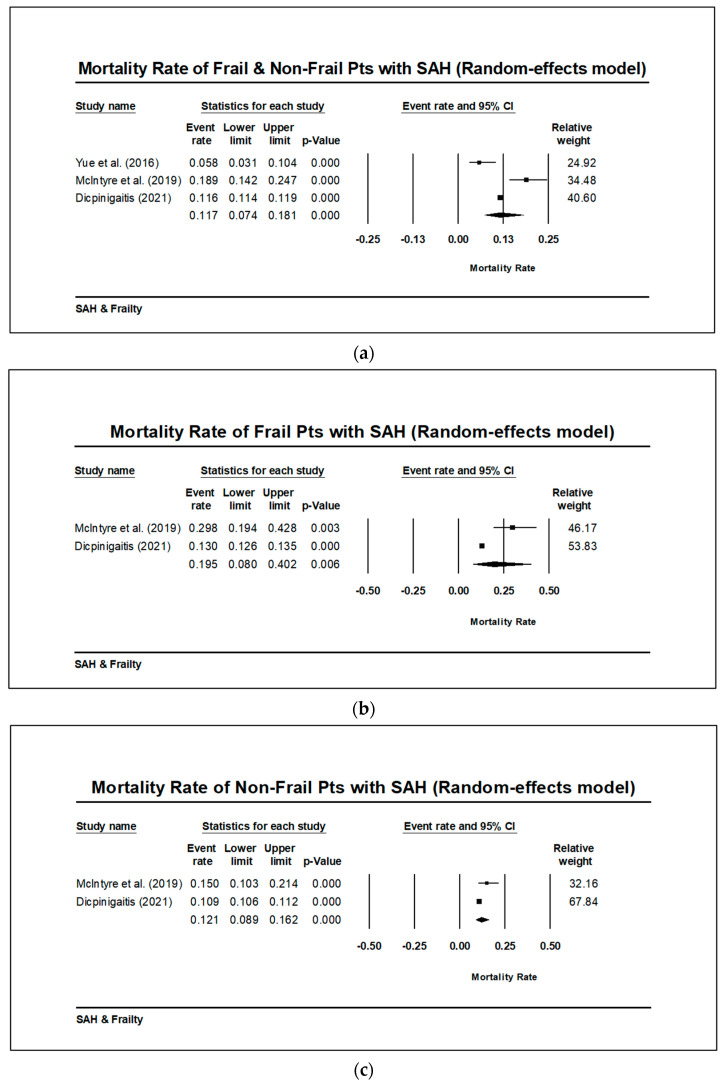
(**a**) Meta-analysis on available data indicates that the overall mortality rate for frail and non-frail patients with aSAH is 11.7% (CI 7.4–18.1%, *p* < 0.001) when using a random-effects model [26,27,32]. (**b**) Meta-analysis on available data indicates that the mortality rate for frail patients with aSAH is 19.5% (CI 8.0–40.2%, *p* < 0.001) when using a random-effects model. Frail patient mortality data was not reported for Yue et al. I^2^ = 0 and Q is not larger than Q (df), suggesting low heterogeneity [26,32]. (**c**) Meta-analysis on available data indicates that the mortality rate for non-frail patients with aSAH is 12.1% (CI 8.9–16.2%, *p* < 0.001) when using a random-effects model. Non-frail patient mortality data was not reported for Yue et al. I^2^ = 0 and Q is not larger than Q (df), suggesting low heterogeneity [26,32].

**Table 1 brainsci-13-01498-t001:** Different frailty measurements. * In some papers, the pre-frail and frail patients are combined for analysis.

Name	Author (Year)	Assessments	Scoring System
Physical Frailty Phenotype (PFP)	Fried et al. (2001) [13]	Weight loss (0 or 1), decreased grip strength (0 or 1), exhaustion (0 or 1), low activity (0 or 1), 10 m walking speed (0 or 1)	0: non-frail 1–2: pre-frail *3–5: frail
Frailty Index (FI)	Mitnitski et al. (2001) [15]	92 total variables that reflect severity of illness or presence of comorbidities	0 or 1 for each selected variable
Modified Frailty Index (mFI)	Velanovich et al. (2013) [14]	11 total variables that focus on accumulated deficits, including history of diabetes mellitus, chronic obstructive pulmonary disease, congestive heart failure, myocardial infarction, history of coronary intervention, hypertension medication, peripheral vascular disease, impaired sensorium, transient ischemic attack or cerebrovascular accident, and cerebrovascular accident with deficit	0 or 1 for each variable
Gill Frailty Measure	Gill et al. (2002) [12]	10 physician-diagnosed chronic conditions and 8 activities of daily living	0 or 1 for each variable

**Table 2 brainsci-13-01498-t002:** Summary of the literature.

Author (Year)	Country	Setting	Study Type (Number of Patients)	Years of Study	Eligible Age (Year)	Patient Age in Years (*n*)	Number of Cases	Level of Evidence
Yue et al. (2016) [27]	China	Single center	Retrospective (109) and Prospective (64)	12/2010 to 12/2013	≥60	categories, 60–69 (134), 70–79 (32), >80 (7)	173	2b
Mclntyre et al. (2019) [26]	USA	Single center	Retrospective	06/2014 to 07/2018	no restriction	mean, 57.6 ± 1.0 range, 14–98	217	2b
Ota et al. (2019) [29]	Japan	Single center	Retrospective	04/2012 to 03/2017	no restriction	mean, 61.6 ± 14	186	2b
Dicpinigaitis et al. (2022) [32]	USA	Multi- center	Retrospective	2010 to 2018	≥18	mean, 55.4 ± 0.1	64,102	2b

## Data Availability

All data used in this meta-analysis is available upon request from the authors. Please contact Fawaz Al-Mufti, MD at fawaz.al-mufti@wmchealth.org for access to the data used in this study.

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
