# Peer review of "Frailty as a Predictor of Outcomes in Subarachnoid Hemorrhage: A Systematic Review and Meta-Analysis"

_brainsci, 2023, doi:10.3390/brainsci13101498_

Round 1

Reviewer 1 Report

Comments and Suggestions for Authors

This is a review on SAH patients and the impact of frailty. While this issue is interesting and relevant to current frailty research my major critics is on why the included studies are limitied to up until 2020? For a study that was submitted end of 2023 this is inapropriate, here are several current studies that were not included:

doi: 10.1007/s00415-023-11805-z

doi: 10.1016/j.jstrokecerebrovasdis.2022.106394

doi: 10.24875/ACM.20000024

doi: 10.1080/02688697.2020

Furthermore, it remains unclear whether this analysis is on aneurysmal SAH or were also traumatic SAH patients included? These cohorts show a complete different pathophysiology and clinical outcome and cannot be mixed in an analysis.

The introduction on SAH should be rewritten: SAH is a neurological/neurosurgical disease.

There is no conservative management of SAH (here again – do we talk about aneurysmal SAH?) its endovascular or neurosurgical. The sentence  “Surgical treatments for SAH entail inherent risks; hence, more conservative management is generally preferred, especially in frailer patients” is simply false.

Figure #1 the grammar correction is activated

The figures are of poor quality and barely readable

Reviewer 2 Report

Comments and Suggestions for Authors

The article is well-written in correct English.

The authors conducted a meta-analysis to evaluate the impact of frailty on Subarachnoid hemorrhage (SAH) outcome. They included 4 studies in the analysis, finding no significant difference in mortality between frail versus non-frail SAH patients; this analysis has been made only on two studies.

Frailty is increasingly gaining importance in surgery, as a means of determining the most appropriate and cost-effective therapeutic strategy in an aging population.   

The authors conduct the study in a rigorous methodological way, according to the PRISMA guidelines for meta-analysis. 

There are obvious limitations (addressed also by the authors). Firstly, the inclusion of only 4 studies leads to a potential publication bias affecting the real mortality rates. Statistical analysis on mortality was made on only two studies and is, therefore, to be interpreted cautiously, especially concerning the heterogeneity found. Finally, all studies included were retrospective and thus classified as low evidence.

The topic is interesting and relatively new. Therefore, this rigorous study can provide new insight into the importance of frailty in the neurosurgical setting.

Reviewer 3 Report

Comments and Suggestions for Authors

This study, through a meta-analysis, evaluates frailty as a predictor of outcomes in subarachnoid hemorrhage. The topic is certainly interesting. With the increase in average age, frailty becomes increasingly current and, therefore, to be evaluated in the management of patient treatment.

In my opinion, the study presents some critical issues:

1) Only 4 studies are evaluated and from a limited number of countries (USA, Japan, China). The data extrapolated and evaluated by the authors cannot reflect those of a wider population nor can they evaluate the reality of other nations (there is no study of European countries). Furthermore, the data does not appear homogeneous (age range considered, variables considered in the definition of "frail patient").

2) The authors do not specify the type of treatment these patients had. SAH "sine materia" requires conservative treatment (monitoring); the presence of aneurysms can direct the patient towards surgical treatment or endovascular treatment. The specific type of treatment will have a different impact on the patient. Consequently, the fragility could interfere with correct outcome. These data do not appear to be reported in the text.

Reviewer 4 Report

Comments and Suggestions for Authors

Thank you very much for the opportunity to review this very interesting article.

In their work, the authors conduct a systematic review focusing on the association of frailty with outcome following subarachnoid hemorrhage.

Overall, it is a clear and concise article, addressing a key topic of neurosurgical practice. Nevertheless, some issues regarding the methodology needs to be addressed.

Please indicate whether all non-traumatic causes of SAH are included or if further limitations have been applied.

Dicpinigaitis et al with over 64000 patients has a low relative weight compared to the expected.

In the relevant figures, negative values in all parameters under investigation are counter-intuitive.

When comparing mortality rates, I would suggest using the odds ratio with values over 1 favoring the one condition and below 1 favoring the other.

Risk for vasospasm and therefore post-hemorrhagic outcome is known to be associated with Fisher score, as the authos also point out. Probably it would be a good idea to include this as a cofactor.

Round 2

Reviewer 3 Report

Comments and Suggestions for Authors

The authors responded comprehensively to my comments